# CATASAN Is a New Anti-Biofilm Agent Produced by the Marine Antarctic Bacterium *Psychrobacter* sp. TAE2020

**DOI:** 10.3390/md20120747

**Published:** 2022-11-27

**Authors:** Caterina D’Angelo, Angela Casillo, Chiara Melchiorre, Concetta Lauro, Maria Michela Corsaro, Andrea Carpentieri, Maria Luisa Tutino, Ermenegilda Parrilli

**Affiliations:** 1Department of Chemical Sciences, University of Naples “Federico II”, Complesso Universitario Monte S. Angelo, Via Cintia 4, 80126 Naples, Italy; 2Istituto Nazionale Biostrutture e Biosistemi—I.N.B.B., Viale Medaglie d’Oro, 305-00136 Rome, Italy

**Keywords:** anti-biofilm, anti-adhesive, bioemulsifier, biosurfactant, *Psychrobacter sp.* TAE2020

## Abstract

The development of new approaches to prevent microbial surface adhesion and biofilm formation is an emerging need following the growing understanding of the impact of biofilm-related infections on human health. *Staphylococcus epidermidis*, with its ability to form biofilm and colonize biomaterials, represents the most frequent causative agent involved in infections of medical devices. In the research of new anti-biofilm agents against *S. epidermidis* biofilm, Antarctic marine bacteria represent an untapped reservoir of biodiversity. In the present study, the attention was focused on *Psychrobacter sp*. TAE2020, an Antarctic marine bacterium that produces molecules able to impair the initial attachment of *S. epidermidis* strains to the polystyrene surface. The setup of suitable purification protocols allowed the identification by NMR spectroscopy and LC-MS/MS analysis of a protein–polysaccharide complex named CATASAN. This complex proved to be a very effective anti-biofilm agent. Indeed, it not only interferes with cell surface attachment, but also prevents biofilm formation and affects the mature biofilm matrix structure of *S. epidermidis*. Moreover, CATASAN is endowed with a good emulsification activity in a wide range of pH and temperature. Therefore, its use can be easily extended to different biotechnological applications.

## 1. Introduction

Biofilms are multicellular microbial aggregates surrounded by a self-producing extracellular polymeric substance (EPS) and they are the most successful and widely distributed form of life on earth [1]. This microbial way of life has a considerable impact on human healthcare since cells in biofilm are often more tolerant and resistant to antibiotics and other antimicrobial agents compared with planktonic cells. The underlying mechanism is the quenching of the activity of antimicrobial substances that diffuse through the biofilm [2], in a form of inhibition identified as diffusion-reaction inhibition [3]. In addition, slow growth within the biofilm and the generation of dormant persister cells further increase the survival of cells to antibiotic actions. Moreover, sessile cells embedded in a biofilm matrix are more tolerant to the host immune system than their planktonic counterpart. Therefore, the classical therapies sometimes cannot achieve the desired results [4]. The US National Institutes of Health stated that over 80% of microbial infections in the body are due to biofilms, many of which are resilient to traditional anti-microbial treatments and some surgery is required to eradicate chronic infections. Therefore, the peculiar properties of pathogens growing in biofilm represent a huge health threat and a rising economic burden [5]. It is compelling to find new approaches for the prevention and treatment of microbial adhesion and biofilm formation in medical settings [6]. 

A biofilm is not simply a structured assemblage of cells, but a dynamic complex system that evolves through a tightly controlled multistep process [7]. Therefore, anti-biofilm strategies can have as targets different stages of biofilm development [8]. All the different stages of biofilm development may be targets of anti-biofilm strategies [8]. For instance, the inhibition of the initial attachment of bacteria to surfaces reduces the chances of biofilm development. Other potential targets are the biofilm formation process as well as the destabilization of the mature biofilm. Consequently, the different anti-biofilm agents are effective using a variety of strategies, as each of them displays distinct chemical-biological properties. Anti-biofilm agents could be EPS synthesis inhibitors or EPS degrading enzymes, adhesion inhibitors, quorum sensing interferents, and, of course, they could have a different origin, being synthetic molecules or nature-derived bioactive compounds [6,8]. 

In the research of new anti-biofilm agents, microorganisms able to thrive in harsh conditions, like in Antarctica, represent an untapped reservoir of biodiversity [9]. Antarctic marine bacteria developed different survival strategies to live in extreme environmental conditions. Some of these are aimed at reducing the presence of competitive microorganisms. Such behavior is necessary when nutrients are limited or difficult to uptake. Biofilm formation allows cells to grow even in highly oligotrophic environments [10,11], and the production of anti-biofilm molecules can reduce the biofilm formation of competitors and their ability to survive. Therefore, is not a surprise that some recent papers report that marine Antarctic bacteria produce and secrete anti-biofilm molecules [12,13,14,15].

Recently, our interest was attracted by *Psychrobacter sp.* TAE2020, an aerobic ƴ-proteobacterium isolated from an Antarctic coastal seawater sample [16]. This marine bacterium is not only able to produce and secrete surfactants and bioemulsifier [16], but its cell-free supernatant strongly affected some specific virulence features of *P. aeruginosa* isolates from CF patients [17].

In this work, we decided to investigate the capabilities of this Antarctic marine bacterium to produce and secrete anti-biofilm compounds active against *Staphylococcus epidermidis* [18]. This human opportunistic pathogen is involved in the infection of any type of indwelling medical devices. Indeed, medical devices, such as intravenous catheters, prosthetic joints, and heart valves provide an opportunity for *S. epidermidis* to gain access to the body and cause infections [19,20,21]. Generally, the success of *S. epidermidis* as a pathogen is attributed to its ability to adhere to both biotic and abiotic surfaces and to form a resistant biofilm, and its remarkable ability to form biofilms is widely regarded as its major pathogenic determinant [22].

## 2. Results

### 2.1. Psychrobacter sp. TAE2020 Produces Anti-Biofilm and Anti-Adhesive Molecules Active against S. epidermidis Biofilm

Preliminary experiments were carried out to assess the ability of *Psychrobacter sp*. TAE2020 to produce anti-biofilm compounds active against different stages of *S. epidermidis* biofilm formation. The anti-adhesive and anti-biofilm activities of *Psychrobacter sp.* TAE2020 cell-free supernatant, named SN_TAE2020, were tested against the reference strain *S. epidermidis* RP62A [23] and against *S. epidermidis* O-47 an *agr*-mutant considered a strong biofilm producer [24,25]. A surface coating assay was performed to assess the ability of SN_TAE2020 to impair *S. epidermidis* surface adhesion, and in particular, the ability of the coated surfaces to avoid biofilm formation by *S. epidermidis* RP62A and *S. epidermidis* O-47 was tested (Figure 1A).

The capability of SN_TAE2020 to inhibit the biofilm formation of *S. epidermidis* was evaluated by adding the supernatant to the *S. epidermidis* growth medium at the beginning of the cultivation (Figure 1B), while the activity on mature biofilm was explored by adding the SN_TAE2020 after biofilm formation (Figure 1C).

As reported in Figure 1A, SN_TAE2020 had anti-adhesive activity against *S. epidermidis* strains and proved able to impair the biofilm formation (Figure 1B) and promote the detachment of staphylococci mature biofilm (Figure 1C).

To assess if the compound/s that inhibit the biofilm formation had a proteinaceous nature, SN_TAE2020 was incubated with proteinase K, and the effect of treated cell-free supernatant was evaluated on *S. epidermidis* O-47 and *S. epidermidis* RP62A biofilm. As shown in Figure 1D, the supernatant treated with proteinase K partially lost the biofilm-inhibiting activity. This result suggested two possible hypotheses. One is that *Psychrobacter sp.* TAE2020 produces more than one anti-biofilm molecule with different chemical features. Alternatively, the produced anti-biofilm molecule/s are very resistant to proteinase K treatment. To clarify this point, SN_TAE2020 was subjected to a liquid-liquid extraction using ethyl acetate as an extraction solvent. The extract obtained by this method is enriched in small organic metabolites. Then, the activity of the obtained extracellular extract (ExEx TAE2020) was evaluated on the biofilm formation of staphylococci. As reported in Figure 1E, ExEx TAE2020 inhibited the biofilm formation of both *S. epidermidis* strains, supporting the presence in SN_TAE2020 of anti-biofilm compounds with different chemical natures: protein/peptide and small organic metabolites.

Furthermore, to define the best condition to produce the molecule/s endowed with the anti-adhesive activity, *Psychrobacter sp.* TAE2020 was grown in planktonic conditions for 24 h, 48 h, 72 h, 96 h, and 120 h at 15 °C. The supernatants were recovered as already described and tested by the surface coating assay. As reported in Figure 2A, the supernatant collected after 72 h of growth, during the stationary phase (Appendix A), showed the strongest anti-adhesive activity against both S. epidermidis strains.

SN_TAE2020 collected after 72 h of growth was concentrated ten-fold by ultrafiltration with membranes of 30kDa MWCO and the anti-adhesive activity of the retentate fraction (SNC_TAE2020) and the permeate fraction (P_TAE2020) was determined by the coating assay against *S. epidermidis* RP62A and *S. epidermidis* O-47. The results demonstrated (Figure 2B) that only SNC_TAE2020 can inhibit the initial attachment of staphylococci cells to the surface, suggesting that the anti-adhesive molecule/s have a molecular weight higher than 30 kDa.

To ascertain more information about the chemical nature of the anti-adhesive compound/s, SNC_TAE2020 was treated with proteinase K or with NaIO_4_ [26], and the anti-adhesive activity of the treated samples was compared to the activity of untreated samples.

Both treatments affected the anti-adhesive activity of SNC_TAE2020 (Figure 2B), indicating that the active anti-adhesive compounds produced by *Psychrobacter sp*. TAE2020 contains peptide bonds and a sugar moiety. Therefore, the active molecules involved in the anti-biofilm activity could be peptides/proteins and/or saccharides. 

### 2.2. Psychrobacter sp. TAE2020 Supernatant Has a Surfactant and Emulsifier Activity 

In previous work, it was demonstrated that *Psychrobacter sp.* TAE2020 cell-free supernatant (SN_TAE2020) has surfactant and emulsifier capability [16]. 

To evaluate the best condition for bio-emulsifier production, the emulsification index determination was performed on SN_TAE2020 collected at different growth times (24 h, 48 h, 72 h, 96 h, and 120 h). The samples were tested in the presence of 2 mL of Dectol by measuring the emulsification index after 24 h (E_24_). As shown in Figure 3A, the SN_TAE2020 collected after 72 h of growth has a higher emulsifying activity. It is remarkable to note that the emulsion is stable for more than 30 days at room temperature (data not shown). This sample was chosen for further characterization experiments. The effects of pH and temperature on SN_TAE2020 bio-emulsifier activity were explored. The emulsification capacity was retained from pH 6 to 12, but it was compromised at pH 3 (Figure 3B). Interestingly, as shown in Figure 3C, the emulsifying activity of SN_TAE2020 is also stable at high temperatures. 

As bio-emulsifiers generally have a molecular weight higher than biosurfactants, we evaluated the surfactant and emulsification properties of SNC_TAE2020 and P_TAE2020. To test if SNC_TAE2020 and P_TAE2020 were able to reduce the surface tension of an aqueous solution, the Du Nouy ring method was applied and the surface tension value for SNC_TAE2020 (53,81 (±0.74) mNm^−1^) and P_TAE2020 (46,75 (±0.35) mN m^−1^) resulted lower than that of Gut medium (71,00 (±0.29) mN m^−1^). The analysis showed that P_TAE2020 was the most active sample, suggesting that the greater contribution to the SN_TAE2020 surfactant properties is due to molecules with a molecular weight lower than 30kDa. Instead, the analysis of the emulsification ability of SNC_TAE2020 and P_TAE2020 showed that P_TAE2020 had no emulsifying activity (Figure 3D). Therefore, emulsification activity is associated with molecules with a molecular weight equal to or higher than 30kDa. 

To ascertain information about the chemical nature of emulsifier compound/s, SNC_TAE2020 was treated with proteinase K or NaIO_4_ (Proteinase K treatment hydrolyses proteins while NaIO_4_ is a gentle oxidizing agent that cleaves cis-diols in carbohydrate sugars). Then, the E_24_ value of treated SNC_TAE2020 was determined, and it resulted to be 85% for the sample treated with proteinase K and 45% for the sample treated with NaIO_4_ (Figure 3E). These data demonstrated that SNC_TAE2020 treated with proteinase K did not lose emulsifying capability, although the treated sample formed less stable emulsification (data not shown), whereas NaIO_4_ treatment reduced the activity of the sample. These results suggested that the bio-emulsifying activity could be due to a carbohydrate molecule.

### 2.3. Purification of Anti-Adhesive and Bioemulsifying Molecules 

A scale-up of the planktonic growth in a bioreactor was set up to accumulate the amount of culture supernatant sufficient for the purification of the anti-adhesive and bio-emulsifying molecules. *Psychrobacter sp.* TAE2020 was grown in a 3 L stirred tank fermenter at 15 °C in Gut medium. The growth was followed for 72 h till the stationary phase, and the supernatant (SN_TAE2020) was collected, separated from the cells, and ultrafiltered as described above.

Two parallel purification protocols were developed to purify the active molecules. 

In the first purification protocol, the SNC_TAE2020 was dialyzed against Milli-Q water by a 30kDa PES membrane, and the dialyzed sample, named D30, was tested for its anti-adhesive and emulsification activity. The assays (Figure 4A,B) showed that D30 has anti-adhesive and emulsification properties. Given the putative proteinaceous nature of the anti-adhesive molecule/s, the D30 sample was analyzed by SDS-PAGE in comparison to SNC_TAE2020, highlighting that D30 displays a protein profile similar to SNC_TAE2020 but with some enriched bands (Figure 4C). The sample D30 was fractionated by gel filtration chromatography and all the collected fractions were assayed for anti-adhesive and emulsifying activities (data not shown). The fraction C2, corresponding to the void volume of the size exclusion chromatography (higher than 350 kDa), had anti-adhesive and emulsification activities (Figure 4D,E). Although diluted, this fraction was analyzed to evaluate the presence of proteins and sugars through electrophoresis analysis (Figure 4F) and DOC-PAGE (Figure 4G). 

In the second purification protocol, SNC_TAE2020 was subjected to adsorption chromatography on Amberlite XAD-2, a polystyrene resin, exploiting the reported ability of the anti-adhesive molecules to adhere to this material [27]. Upon SNC_TAE2020 loading, the column was extensively washed with the Gut medium, while the elution steps were performed with methanol or 0.5 M NaOH in methanol. The collected chromato-graphic fractions were tested by the surface coating assay to evaluate their anti-adhesive proprieties against staphylococci (Figure 5A). The fractions eluted with methanol not only display a strong anti-adhesive activity against both staphylococcal strains (data not shown), but also interfere with the bacterium adhesion on the entire well surface (Figure 5A). When the MeOH fraction was diluted four times and tested at a lower concentration, the usual anti-adhesive activity was observed (Figure 5B). 

The SDS-PAGE electrophoresis analysis demonstrated that the methanol fraction MeOH has a less complex protein profile with respect to the unbound and wash fractions, with some quite enriched protein bands (Figure 5C). It is interesting to note that the MeOH fraction also presents good emulsification activity (E_24_ 70%). 

### 2.4. Catasan Identification

To identify the molecule/s responsible for the anti-adhesive activity, the active fractions obtained by applying the two different purification protocols were compared by SDS-PAGE. Figure 6A shows that some protein bands are present in all the analyzed functions (i.e., the C2 fraction, D30, and MeOH fraction), suggesting that some of these proteins could be responsible for the anti-adhesive activity.

The protein with a molecular weight of about 40kDa, common to all the fractions, was selected, in-situ hydrolyzed, and identified by LC-MSMS. The resulting data led to the identification of a *Psychrobacter sp*. TAE2020 protein named PsyOmp38. This protein displays a good similarity to the OmpA protein family from different *Psychrobacter* strains (data not shown). An in-silico analysis of the amino acid sequence allowed the identification of a putative signal peptide and the presence of several beta-barrel regions predicted to be internal to the outer membrane (data not shown).

A protein homologous to PsyOmp38 is a key constituent of Alasan, a bioemulsifier produced by *Acinetobacter radioresistens* KA53 [28]. Alasan is a complex of proteins and polysaccharides, and the major emulsification activity of this complex is associated with a 35.77 kDa protein named AlnA [29]. The sequence of PsyOmp38 was aligned (Clustal W) with different OmpA-like proteins of Acinetobacter strains endowed with emulsification activity [30] and, as shown in Figure 6C, PsyOmp38 shares four hydrophobic loops (HL), previously reported as crucial for this activity, with the other aligned proteins [29]. 

The purified C2 fraction was analyzed by NMR spectroscopy and chemical analyses. The ^1^H NMR spectrum of the fraction C2 (Figure 7A) is suggestive of the presence of a polysaccharide since the main signals were identified in the regions 5.3–4.5 ppm (anomeric signals), 4.5–3.5 ppm (carbinolic signals), 2.1–1.8 ppm (N-acetyl groups), and 1.5–1.0 ppm (methyl groups). In addition, the N-acetyl groups indicated that amino sugars are included in the structure. The DOC-PAGE analysis of the C2 fraction, visualized after Alcian Blue and silver nitrate, confirmed the presence of bands at high molecular weight attributable to a polysaccharide, together with a fast-migrating band due to the presence of the lipopolysaccharide (LPS) (Figure 7B). The glycosyl and fatty acid composition was obtained after derivatization into acetylated methyl glycosides (AMG) and fatty acid methyl esters (FAME), respectively. GC-MS chromatogram of AMG disclosed mainly the presence of galactosamine (GalN), confirming the presence of N-acetyl groups as suggested by the NMR analysis. Furthermore, glucosamine (GlcN) and 3-deoxy-D-manno-octulosonic acid (Kdo), a peculiar monosaccharide of the LPS, were detected. Finally, the GC-MS chromatogram of FAME indicated the presence of C12:0–3OH of LPS, and signals attributable to C14:0, C15:0, C16:1, C16:0, C18:1, and C18:0.

Due to the similarities between our anti-adhesive and bioemulsifier molecule and Alasan, the new active compound was named CATASAN (Cold AdapTed alASAN).

The purified CATASAN showed strong anti-adhesive (data not shown) and anti-biofilm activity (Figure 8A) against *S. epidermidis* strains at a concentration of 100 µg mL^−1^. At a concentration of 1 mg mL^−1^, it promotes the detachment of staphylococci mature biofilm (Figure 8D) and presents good emulsification activity (E_24_ 72%).

The CATASAN effect on *S. epidermidis* O-47 and *S. epidermidis* RP62A biofilms was further investigated by confocal laser scanning microscopy (CLSM), to analyze the biofilm structure and cell integrity by LIVE/DEAD^®^ Biofilm Viability Kit (Figure 8B,E). As shown, the CLMS analysis confirmed that the CATASAN has the capability to reduce *S. epidermidis* biofilm formation (Figure 8B) without affecting cell viability and to promote the detachment of staphylococci mature biofilm (Figure 8E). The CLSM image stack data were further analyzed using the COMSTAT image analysis software package [31] to evaluate the different variables describing the biofilm structure. As expected, the values of the biomass and the average thickness of the biofilm obtained in the presence of CATASAN were lower if compared to the values obtained without the CATASAN, while an increased roughness coefficient is observed for the treated sample (Figure 8C,F). This dimensionless factor provides a measure of the thickness variation of a biofilm, and thus it is used as an indirect indicator of biofilm heterogeneity. The assays revealed that the treatment resulted in an unstructured biofilm.

## 3. Discussion

Preliminary data demonstrated that the Antarctic marine bacterium *Psychrobacter sp.* TAE2020 can secrete molecules endowed with surfactant and emulsifying properties [16] and it is able to interfere virulence of several *Pseudomonas aeruginosa* clinical isolates [17]. In this paper, we focused our attention on its ability to affect *S. epidermidis* biofilm formation. Our experimental work started with the assessment of anti-biofilm activity of *Psychrobacter sp.* TAE2020 cell-free supernatant against two S. epidermidis strains. Surprisingly, *Psychrobacter sp*. TAE2020 cell-free supernatant proved able to interfere with the surface adhesion, with biofilm formation, and its presence promoted the detachment of mature biofilm. Such different activities suggested the presence of different molecules involved in, or the presence of a single, very effective, anti-biofilm molecule. 

We collected several physico-chemical information on the molecule/s involved in the inhibition of attachment to set up a suitable purification strategy. The results demonstrated that the anti-adhesive molecule is produced by the cell during all growth phases, but higher activity is recorded after 72 h of growth. The compound/s has a molecular weight higher than 30 kDa and the anti-adhesive activity is completely abolished by treatment with protease K or with NaIO_4_, suggesting the involvement of protein and/or carbohydrates in the described activity. 

Since several works report that biosurfactants [32] and bioemulsifier [33] have anti-biofilm activity, we explored if the sample endowed with anti-adhesive activity also works as a bioemulsifier or biosurfactant. Although bio-emulsifiers and biosurfactants are both amphiphilic, they are chemically different. Biosurfactants are generally low molecular weight molecules known for their surface activity, which reduces the interfacial tension between different phases. Moreover, they can form stable emulsions [34,35]. Bio-emulsifiers [36] are high-weight molecular biopolymers, e.g., heteropolysaccharides, lipopolysaccharides, lipoproteins, and mixtures of these components. They can bind tightly to hydrocarbons and oil forming a barrier that prevents drop coalescence. Thus, they show, like biosurfactants, an outstanding capability to stabilize emulsions, but they do not have any effects on the solution’s surface tension [36] 

The analysis performed on the supernatant fractions SNC_TAE2020 and P_TAE2020 demonstrated that the surfactant activity was mainly related to low molecular weight molecule/s while the anti-adhesive and the emulsification activities proved to be associated with higher molecular weight molecules. Interestingly, the treatment of SNC_TAE2020 with proteinase K did not interfere with emulsification activity while the treatment with NaIO_4_ abolished the ability of the sample to stabilize emulsification. The latter results supported our hypothesis that the molecules involved in anti-adhesive and emulsification activities, although both present in the fraction SNC_TAE2020, were different. Therefore, we set up two parallel strategies to purify the active molecules.

One approach was based on the ability of the anti-adhesive molecule/s to bind polystyrene and, as expected, the fraction eluted from the polystyrene chromatographic resin with methanol (MeOH) had the ability to interfere with *S. epidermidis* adhesion to polystyrene. Surprisingly, it presented good emulsification activity. 

The alternative purification procedure was inspired by the protocols designed to purify emulsifiers produced by marine bacteria. Indeed, as reported in the literature, some emulsifiers produced by marine bacteria in low ionic strength solutions tend to form aggregates of high molecular weight, making their recovery possible through a membrane of appropriate cut-off [37,38]. The dialysis of SNC_TAE2020 against water was aimed to trigger the aggregation of the bio-emulsifiers but surprisingly we recovered the activity not in the insoluble fraction but in the dialyzed solution, proving that the active molecule/s was soluble in distilled water. The dialyzed SNC (D30) resulted to be able to stabilize oil in water emulsions but also to work as an anti-adhesive compound. The next step consisted of the fractionation of the D30 sample by gel filtration chromatography, the fractions obtained were assayed by coating and emulsification assay. The most active fraction (C2) was analyzed by electrophoresis analysis in comparison with MeOH and D30 fractions. The direct comparison of the three different active fractions showed that C2 had mainly a protein band in common with D30 and MeOH samples and staining with silver nitrate on the DOC-PAGE highlighted the presence of sugars in all active fractions. The analysis by mass-spectrometry revealed that the protein present in all active fractions was TAE2020_00012 Outer membrane protein PsyOmp38. 

The PsyOmp38 proved to be homologous to several outer membrane proteins of the OmpA family commonly found in Gram-bacteria. The function of Omp proteins is related to osmosis, bacterial conjugation, and signal transmission, and in some species, OmpA proteins work as a secreted emulsifier [39]. This is the case of the OmpA identified in *A. radioresistens* K53, named AlnA, that is a component of a complex of proteins and sugars named Alasan [40]. The AlnA protein is reported to be endowed with emulsification activity, while the other components are supposed to be relevant for the stability and secretion of the complex [28]. As with other OmpA proteins displaying emulsifying properties, AlnA has four putative extra-membrane loops that are highly hydrophobic and essential for emulsifying activity [29,39]. *Psycrobacter sp.* TAE2020 Omp38 has 46% of identity with *A. radioresistens* K53 AlnA. Interestingly, the alignment of PsyOmp38 with other OmpA-like proteins with emulsification activity [29,30,39] revealed the presence of four highly conserved hydro-phobic regions. 

In the case of the Alasan complex, the carbohydrate component consists of a polysaccharide containing D-glucose, D-galactose, N-Acetyl-D-glucosamine, N-Acetyl-D-galactosamine, and decorated by covalently bound alanine [41]. The polysaccharide moiety of another emulsifier named Emulsan [42] isolated from Acinetobacter sp. ATCC 31012 (RAG-1) (later renamed *Acinetobacter venetianus* RAG-1) contains D-galactosamine, D-galactosaminuronic acid, and di-amino-6-deoxy-D-glucose. It was demonstrated that this complex contains approximately 80% (*w/w*) lipopolysaccharide (LPS) and 20% (*w/w*) high molecular weight exopolysaccharide (EPS) and that some physical properties of Emulsan can be attributed to the LPS fraction while the EPS is responsible for the emulsifying activity [43]. In agreement with the previous studies, our results demonstrated that CATASAN complex contains a mixture of LPS and a high molecular weight polysaccharide. The GC-MS analysis of acetylated methyl glycosides revealed the presence of galactosamine as the main component. The 1H NMR spectrum clearly indicated the presence of many acetyl groups not immediately explainable with the occurrence of the sole galactosamine residue. Further analyses will be necessary to obtain the detailed structure of the polysaccharide component of CATASAN and to characterize its interaction with the polypeptide chains. Usually, the emulsifying activity is related to molecules’ amphipathic properties. In the case of Emulsan from *A. venetianus* RAG-1, this feature is due to the presence of fatty acids and hydrophilic polysaccharides [42]. In Emulsan from *A. calcoaceticus* BD4, its amphipathic properties derive from the association of an anionic hydrophilic polysaccharide with proteins [44]. In the Alasan, the emulsifying activity is mainly related to AlnA protein [29], although AlnA alone is not capable of producing stable emulsion [40]. In the case of CATASAN, the polysaccharidic component is crucial for the emulsification activity, though the CATASAN treated with NaIO_4_ showed a residual emulsification activity and the sample treated with proteinase K formed a less stable emulsification. Future structural and functional studies will clarify the role of protein and polysaccharidic components in emulsifying activity. In any case, the CATASAN properties allow us to include it in the bioemulsifer family [38,45,46]. 

Even though several structural-functional aspects of CATASAN must be clarified and it is easy to explain its emulsification activity, the mechanisms by which the complex is able to prevent biofilm formation, disturb surface attachment, and affect mature biofilm matrix structure are simply not thinkable.

The modification of the physico-chemical properties [47,48] of the surface (e.g., surface charge, hydrophobicity, surface free energy), in the presence of CATASAN, could be responsible for the reduced microbial adhesion to a polystyrene surface. Although very initial data indicate that CATASAN could reduce the hydrophobicity of the polystyrene surface (Appendix A) and previous studies have demonstrated that surface hydrophobicity influences the initial binding of S. epidermidis [49], it is likely that the reduction of hydrophobicity is not the only reason of the reduction of cell adhesion. Indeed, the surface charge also plays a key role in the first attachment of *S. epidermidis* to a solid surface [50]. The inhibition of the first attachment is not automatically related to the extent of biofilm formed. It was demonstrated that high levels of initial adherence in *S. epidermidis* do not necessarily lead to thick biofilm formation and vice versa [49]. These two steps of the biofilm cycle are regulated by different molecular mechanisms. The ability of CATASAN to reduce biofilm formation and to favor mature biofilm detach could be related to the ability of the molecule to interfere with the formation and the stability of the *S. epidermidis* biofilm matrix. The concentration of CATASAN required for the mature biofilm detach is not suggestive of a QS-mediated action. Therefore, the anti-biofilm properties could be related to CATASAN emulsification capacity. Indeed, several reports describe the anti-biofilm activity of amphipathic molecules [35,51,52]. Interestingly, in the case of *S. epidermidis*, a group of peptides with surfactant properties (the phenol-soluble modulins (PSMs) [53] ) is involved in the dispersion stage. Modulins disrupt non-covalent bonds formed between cells and matrix during biofilm development to promote the formation of channels and biofilm detachment [54]. In agreement with this hypothesis, the CLSM analyses on *S. epidermidis* treated biofilm revealed that the CATASAN action not only reduces the biofilm biomass without affecting cell viability but deeply modifies the *S. epidermidis* biofilm structure. 

## 4. Conclusions

In conclusion, although the structural-functional properties of CATASAN remain to be clarified, our data support its impressive anti-biofilm activity and its emulsification activity in a wide range of pH and temperature, making its exploitation in several biotechnological applications worthy of further investigation. Furthermore, the capability of the Antarctic marine bacterium *Psycrobacter sp*. TAE2020 to produce a multiplicity of bioactive compounds, not only CATASAN, but also surfactants [16] and low molecular molecules active against *P. aeruginosa* [17], confirms the potential of marine cold-adapted bacteria as a source of novel and interesting bioactivities to be exploited towards the development of innovative approaches for the prevention and treatment of biofilm-associated infections.

## 5. Materials and Methods

### 5.1. Bacterial Strains and Culture Conditions

Bacterial strains used in this work were *Psychrobacter sp*. TAE2020, collected in 1992 from seawater near French Antarctic Station Dumont d’Urville, Terre Adélie (66°40′ S; 140° 01′ E); S. *epidermidis* O-47 isolated from clinical septic arthritis and kindly provided by Prof. Gotz [55]. *S. epidermidis* RP62A reference strain was isolated from an infected catheter (ATCC collection no. 35984). 

*Psychrobacter sp*. TAE2020 was grown in synthetic medium Gut (L-Glutamic acid 10 g L^−1^, NaCl 10 g L^−1^; NH_4_NO_3_ 1 g L^−1^; KH_2_PO·7H_2_O 1 g L^−1^; MgSO_4_·7H_2_O 200 mg L^−1^; FeSO_4_·7H_2_O 5 mg L^−1^; CaCl_2_·2H_2_O 5 mg L^−1^) [56] in planktonic conditions at 15 °C under vigorous agitation (250 rpm) for different times of growth (24 h, 48 h, 72 h, 96 h, 120 h). The cell-free supernatant (SN_TAE2020) was recovered by centrifugation at 7000 rpm at 4 °C for 30 min, sterilized by filtration through membranes with a pore diameter of 0.22 μm, and stored at 4 °C until use. 

Staphylococci were grown at 37 °C in Brain Heart Infusion broth (BHI, Oxoid, UK), and biofilm formation was assessed in static conditions while planktonic cultures were performed under agitation (180 rpm).

All strains were maintained at −80 °C in cryovials with 20% of glycerol. 

### 5.2. Anti-Adhesive Assay

*Surface coating assay*. A volume of 5 μL of the tested sample or Gut (as control) was deposited onto the center of a well of a 24-well tissue-culture-treated polystyrene microtiter plate. The plate was incubated at room temperature to allow complete evaporation of the liquid in sterile conditions. The wells were then filled with *S. epidermidis* RP62A or *S. epidermidis* O-47 cultures in exponential growth phase diluted in BHI with a final concentration of about 0.1 and 0.001 OD_600nm_ respectively and incubated at 37 °C in static condition. After 24 h, wells were rinsed with water and stained with 1 mL of 0.1% crystal violet. Stained biofilms were rinsed with water and dried, and the wells were photographed.

### 5.3. Biosurfactant Assays

*Drop-collapse test* Drop collapse assay was performed by the deposition of 50 µL droplets of water on a hydrophobic (polystyrene) surface coated with CATASAN (1 mg mL^−1^). Methylene blue was added to stain the samples for photographic purposes and did not influence the shape of the droplets. The spreading of the droplet on the surface was observed after 5 min.

*Emulsification activity (E_24_).* The emulsification index (E_24_) [57] was determined by mixing 2 mL of Dectol (decane and toluene 65:35 v/v) with 1 mL of the test sample in 5 mL glass vials, vortexed at maximum speed using the IKA T-10 Basic Ultra Turrax Homogenizer IKA-Werke GmbH, Staufen, Germany for 3 min, and allowed to stand for 24 h. The emulsification index, E_24_, was determined by calculating the ratio between the height of emulsifying layer and the total height, multiplied by 100.

*Surface tension measurement.* The surface tension, γ, was measured through De Nouy ring method using a KSV Sigma 70 digital tensiometer (Dyne Testing Ltd., NewtonHouse, Lichfield, UK) equipped with an automatic device to set the time between two consecutive measurements and to select the rising velocity of the platinum ring [58]. The ring rising velocity was set low enough to reach the equilibrium between the air–solution interface and the solution bulk. At least three surface tension measurements were performed on a 10 mL of sample.

### 5.4. Physico-Chemical Properties of Active Compound/S

*Proteinase K treatment.* To analyze the physico-chemical properties of active compound/s, proteinase K (Sigma Aldrich, St Louis, MO) was added to the sample at a final concentration of 2 mg mL^−1^ and the reaction was incubated for 2 h at 37 °C. As controls, the sample was incubated without proteinase K for 2 h at 37 °C which did not impair the activities.

*Sodium periodate treatment.* For the polysaccharide treatment, NaIO_4_ (Sigma Aldrich, St Louis, MO, USA) at a final concentration of 20 mM was added to the sample and the reaction was incubated for 24 h at 37 °C. As controls, the same treatment was performed on Gut medium to exclude an effect due to the NaIO_4_. Sodium periodate (NaIO_4_) is a strong oxidizing agent mainly used for the oxidative cleavage of 1,2-diols (vicinal diols).

### 5.5. Studies of the Bioemulsifier Stability 

The effect of temperature on the biosurfactants’ stability was evaluated by heating the SN_TAE2020 samples in a water bath for 30 min at 37 °C, 60 °C, 80 °C, and 100 °C. The samples were cooled to room temperature. After the designated times, the emulsification activity was measured for each sample to check for any possible changes.

Analysis of pH stability was performed by adjusting the pH of the supernatant to 3, 6, 9, and 12. The emulsification activity was then measured as described above 

### 5.6. SNC_TAE2020 and P_TAE2020 Preparation 

SN_TAE2020 was concentrated 10-fold with Amicon Ultrafiltration cell equipped with a 30 kDa cut-off PES Millipore Ultrafiltration Disc (Merck KGaA, Darmstadt, Germany). Then, retentate fraction (SNC_TAE2020) and permeate fraction (P_TAE2020) were collected.

### 5.7. Preliminary Purification of the Active-Compounds 

*Adsorption chromatography*. The primary enrichment of the active compound/s was achieved by adsorption chromatography on a polystyrene resin (Amberlite XAD-2; Rohm and Haas, Philadelphia, Pa.). The resin (4 g) was placed in a glass column (11 cm by 1 cm). The column was equilibrated Gut medium and then 40 mL of retentate fraction (SNC_TAE2020) was applied at a flow rate of approximately 1 mL min^−1^. The column was then washed with 3-bed volumes of Gut. The elution of the active compound/s was subsequently carried out with methanol and NaOH (0.5M) in methanol. Fractions of 30 mL were collected. The unbound fractions and the fractions eluted during the washing were concentrated (ten-times), while those eluted with methanol and NaOH in methanol were recovered, dried, to obtain more concentrated samples, and resuspended in a small volume (5 mL) of Gut medium. Each chromatographic fraction was analyzed by surface coating assay against *S. epidermidis* O-47 and *S. epidermidis* RP62A.

*Gel filtration.* The SNC_TAE2020 was dialyzed against Milli-Q water using a centricon cut-off 30kDa (Merck KGaA, Darmstadt, Germany). After the dialysis, the concentrated supernatant was further purified on a Sephacryl S-300HR (GE Healthcare Life Sciences, 0.5 × 110 cm, flow rate 15.6 mL h^−1^, fraction volume 2.5 mL) eluted with 0.05 M ammonium hydrogen carbonate. 

### 5.8. Large Scale Coultivation of Growth Psycrobacter sp. TAE2020 

*Psycrobacter sp.* TAE2020 bacterial culture was grown in Gut medium in a Stirred Tank Reactor 3 L fermenter (Applikon, Schiedam, The Netherlands) connected to an ADI-1030 Bio Controller with a working volume of 1 L. The bioreactor was equipped with the standard pH, pO2, and level- and temperature sensors for the bioprocess monitoring. The culture was carried out at 15 °C for 72 h in aerobic conditions (30% dissolved oxygen). Supernatant was recovered by centrifugation at 7000 rpm. Then, it was sterilized by filtration through membranes with a pore diameter of 0.22 µm and stored at 4 °C until use. 

### 5.9. SDS PAGE 

Protein samples (prepared in Laemmli buffer 4x followed by boiling at 95 °C for 10 min) were separated on SDS-PAGE gels. The gels were stained with colloidal Coomassie and the protein sizes were determined by comparing the migration of the protein band to a molecular mass standard (Unstained Protein Molecular Weight Marker, Thermo Fisher Scientific Waltham, MA, USA). 

### 5.10. DOC PAGE

DOC-PAGE was performed using the system of Laemmli [59] with sodium deoxycholate (DOC) as the detergent. The separating gel contained final concentrations of 14% acrylamide, 0.1% DOC, and 375 mM Tris/HCl (pH 8.8). The stacking gel contained 4% acrylamide, 0.1% DOC, and 125 mM Tris/HCl (pH 6.8). The electrode buffer was composed of SDS (1 g L^−1^), glycine (14.4 g L^−1^), and Tris (3.0 g L^−1^). The electrophoresis was performed at a constant amperage of 30 mA. The gels were fixed in an aqueous solution of 40% ethanol and 5% acetic acid, then visualized after Alcian Blue and silver nitrate staining for sugar analysis [60] or colloidal Coomassie for protein analysis.

### 5.11. Sugar and Fatty Acids Analyses

Monosaccharides were analyzed as acetylated methyl glycosides, as reported previously [61]. Briefly, 1mg of sample was dried over P_2_O_5_ for 1 h; and the methanolysis was performed in 1 mL of HCl/MeOH 1 M at 80 °C for 20 h. The obtained product was extracted three times with hexane, and the methanol layer was dried and acetylated with 50 µL of Ac_2_O and 50 µL of Pyr at 100 °C for 30 min. 

The hexane layer containing fatty acids methyl esters was also analyzed to obtain fatty acids composition. The samples were analyzed on an Agilent Technologies gas chromatograph 7820A equipped with a mass selective detector 5977B and an HP-5 capillary column (Agilent, 30 m × 0.25 mm i.d.; He as carrier gas). Acetylated methyl glycosides were analyzed using the following temperature program: 140 °C for 3 min, then 140 → 240 °C at 3 °C min ^−1^. Finally, fatty acids methyl esters were analyzed with the following temperature program: 140 °C for 3 min, then 140 → 280 °C at 10 °C min ^−1^, at 280 °C for 20 min.

### 5.12. NMR Spectroscopy

^1^H NMR spectrum was performed in D_2_O at 298 K using a Bruker Avance 600 MHz equipped with a cryoprobe. ^1^H chemical shifts were determined by using acetone as external standard (δ_H_ 2.225 ppm) [62].

### 5.13. Biofilm Inhibiting Assay

The quantification of in vitro biofilm production was based on the method described by Christensen with slight modifications [63]. For staphylococcal biofilm formation in the presence of *Psychrobacter sp*. TAE2020 cell-free supernatant (SN_TAE2020), the wells of a sterile 96-well flat-bottomed polystyrene plate were filled with *S. epidermidis* RP62A or *S. epidermidis* O-47 cultures in exponential growth phase diluted in BHI 2x with a final concentration of about 0.1 and 0.001 OD_600nm_, respectively. Each well was filled with 100 μL of cultures and 100 μL of supernatant. In this way, the supernatant was used diluted 1:2 with a final concentration of 50%.

As control, the first row was filled with 100 μL of cultures and 100 μL of medium used for the growth of TAE2020 (untreated bacteria). The plates were incubated aerobically for 24 h at 37 °C. Biofilm formation was measured using crystal violet staining. After incubation, planktonic cells were gently removed; and wells were washed three times with sterile PBS and thoroughly dried. Each well was then stained with 0.1% crystal violet and incubated for 15 min at room temperature, rinsed twice with double-distilled water, and thoroughly dried. The dye bound to adherent cells was solubilized with 20% (*v/v*) glacial acetic acid and 80% (*v/v*) ethanol. After 30 min of incubation at room temperature, the OD_590nm_ was measured to quantify the total biomass of biofilm formed in each well. Each data point was composed of five independent samples. 

SN_TAE2020 were subjected to proteinase K treatment. The anti-biofilm activity of treated and untreated supernatant was evaluated using the microtiter plate assay against *S. epidermidis* strains as previously described. Each data point was composed of five independent samples.

For the assay with organic extract: dried extract (ExEx TAE2020) was first dissolved in dimethyl-sulfoxide (DMSO), then in the culture medium BHI. Then, 100 uL of *S. epidermidis* RP62A or *S. epidermidis* O-47 cultures in exponential growth phase diluted in BHI with a final concentration of about 0.1 and 0.001 OD_600nm_, respectively and 100 µL of organic extracts (ExEx TAE2020) were added into each well with a final concentration of 1 mg mL^−1^ and a DMSO concentration lower than 4% *v/v*, since this DMSO concentration does not interfere with cell growth and bacteria viability [64]. The anti-biofilm activity of organic extract was evaluated using the microtiter plate assay as previously described. Each data point was composed of three independent samples.

For the assay with CATASAN: the wells of a sterile 96-well flat-bottomed polystyrene plate were filled with 200 μL of *S. epidermidis* RP62A or *S. epidermidis* O-47 cultures in exponential growth phase diluted in BHI with a final concentration of about 0.1 and 0.001 OD_600nm_, respectively. The plate was incubated at 37 °C for 24 h in the absence and in the presence of CATASAN (100 ug mL^−1^). Biofilm formation was measured as previously described. Each data point was composed of four independent samples.

### 5.14. Mature Biofilm Assay

The anti-biofilm activity of cell-free supernatant was also evaluated on preformed biofilm of *S. epidermidis* RP62A or *S. epidermidis* O-47. The wells of a sterile 96-well flat-bottomed polystyrene plate were filled with *S. epidermidis* RP62A or *S. epidermidis* O-47 cultures in exponential growth phase diluted in BHI with a final concentration of approximately 0.1 and 0.001 OD_600nm_, respectively. Each well was filled with 200 μL of cultures. The plates were aerobically incubated for 24 h at 37 °C. After 24 h, the contents of the plates were poured off and the wells were washed with sterile distilled water to remove the unattached bacteria. Hence, 200 μL of the cell-free supernatant (SN_TAE2020) or CATASAN at a concentration of 1 mg mL^−1^ was added into each well. As control, 200 μL of Gut medium was added. The plates prepared in this way were aerobically incubated for an additional 24 h (48 h in total) at 37 °C. After each time-point the plates were analyzed as previously described.

Each data point was composed of five independent samples.

### 5.15. In-Situ Hydrolysis, LC-MS/MS Analysis and Protein Identification

Mono-dimensional SDS-PAGE gel was stained with Coomassie Brilliant Blue, the band approximately at 50 kDa, was excised and de-stained with 100 µL of 0.1 M ammonium bicarbonate (AMBIC) and 130 µL of acetonitrile (ACN) and subsequently subjected to in-situ hydrolysis with 0.1 µg µL^−1^ trypsin mM in AMBIC for 18 h at 37 °C. The hydrolysis was stopped by adding acetonitrile and 0,1% formic acid. The sample was then filtered and dried in a vacuum centrifuge. 

The peptide mixtures thus obtained were directly analyzed by LTQ Orbitrap XL™ Hybrid Ion Trap-Orbitrap Mass Spectrometer (Thermo Fisher Scientific, Bremen, Germany). C-18 reverse phase capillary column 75 μm × 10 cm (Thermo Fisher Scientific) was performed using a flow rate of 300 nL min^−1^, with a gradient from eluentA (0.2% formic acid in 2% acetonitrile) to eluent B (0.2% formic acid in 95% acetonitrile). The following gradient conditions were used: t = 0 min, 5% solvent B; t = 10 min, 5% solvent B; t = 90 MIN, 50% solvent B; t = 100 min, 80% solvent B; t = 105 min, 100% solvent B; t = 115 min, 100% solvent B; t = 120 min; 5% solvent B. Peptide analysis was performed using the data-dependent acquisition of one MS scan followed by CID fragmentation of the five most abundant ions. 

For the MS scans, the scan range was set to 400–1800 m/z at a resolution of 60,000, and the automatic gain control (AGC) target was set to 1 × 106. For the MS/MS scans, the resolution was set to 15,000, the AGC target was set to 1 × 105, the precursor isolation width was 2 Da, and the maximum injection time was set to 500 ms. The CID normalized collision energy was 35%. Data were acquired by Xcalibur™ software (Thermo Fisher Scientific). 

In-house Mascot software (version 2.4.0) was used as a search engine to identify proteins. The TAE2020 (2644 sequences; 878,869 residues) proteins database was used for proteins identification.

The software returns a list of proteins associated with a probability index (score), calculated as −10 × Log P, where P is the probability that the observed event is a random one. Proteins are considered as identified if a minimum number of 2 peptides reach the calculated threshold score.

### 5.16. Organic Extraction Protocol

SN_TAE2020 obtained after 72 h of growth was subjected to a liquid–liquid extraction to obtain the extracellular extract (ExEx TAE2020) without adding cryoprotectants. In detail, it was thawed and stirred with ethyl acetate in a volume ratio of 2:1 (assay percent range ≥ 99.5%) (Sigma-Aldrich, St. Louis, MO, US) and mixed at 1% with formic acid (assay percent range = 90%; JT Baker, Munich, Germany). The solution was stirred for at least 30 min and subsequently centrifuged at 3000 rpm for 30 min. The resulting two phases were separated. The organic phase was recovered and dried using a rotary evaporator, Rotavapor (Buchi R-210, Rodano (MI) Italy), at a temperature lower than 40 °C. The resulting organic extracts (ExEx TAE2020) were aliquoted and stored at −20 °C until the use.

### 5.17. CLSM Analysis

The activity of CATASAN against staphylococcal biofilms was evaluated by Confocal Laser Scanning Microscopy (CLSM). Biofilms were formed on NuncTM Lab-Tek^®^ 8-well Chamber Slides (n◦17744; Thermo Scientific, Ottawa, ON, Canada). Briefly, the wells of the chamber slide were filled with 300 μL of *S. epidermidis* RP62A or *S. epidermidis* O-47 cultures in exponential growth phase diluted in BHI with a final concentration of approximately 0.1 and 0.001 OD_600nm_, respectively. The culture was incubated at 37 °C for 24 h in the absence (control) and in the presence of CATASAN (100 ug mL^−1^) to assess its antibiofilm activity and its influence on cell viability. The biofilm cell viability was determined by the FilmTracer™ LIVE/DEAD^®^ Biofilm Viability Kit (Molecular Probes, Invitrogen, Carlsbad, CA, USA), following the manufacturer’s instructions. After rinsing with filter-sterilized PBS, each well of the chamber slide was filled with 300 µL of working solution of fluorescent stains, containing SYTO^®^9 green-fluorescent nucleic acid stain (10 µM) and propidium iodide, the red-fluorescent nucleic acid stain (60 µM), and incubated for 20–30 min at room temperature, protected from light. All excess stain was removed by rinsing gently with filter-sterilized PBS. All microscopic observations and image acquisitions were performed with a confocal laser scanning microscope (LSM700-Zeiss, Jena, Germany) equipped with an Ar laser (488 nm) and a He-Ne laser (555 nm). Images were obtained using a 20x/0.8 objective. The excitation/emission maxima for these dyes are 480/500 nm for SYTO^®^9 and 490/635 nm for PI. Z-stacks were obtained by driving the microscope to a point just out of focus on both the top and bottom of the biofilms. Images were recorded as a series of tif files with a file-depth of 16 bits. The COMSTAT software package [31] was used to determine biomasses (μm^3^ μm^−2^), average thicknesses (µm), and roughness coefficient (Ra*). For each condition, two independent biofilm samples were used.

The activity of CATASAN on preformed biofilm of *S. epidermidis* RP62A or *S. epidermidis* O-47 was also evaluated. The wells of the chamber slide were filled with *S. epidermidis* RP62A or *S. epidermidis* O-47 cultures in exponential growth phase diluted in BHI with a final concentration of approximately 0.1 and 0.001 OD_600nm_, respectively. Each well was filled with 300 µL of cultures. The plates were aerobically incubated for 24 h at 37 °C. After 24 h, the contents of the plates were poured off and the wells were washed with sterile distilled water to remove the unattached bacteria. Then, 300 µL of the CATASAN at a concentration of 1 mg mL^−1^ was added into each well. As control, 300 µL of Gut medium was added. The plates prepared in this way were aerobically incubated for an additional 24 h (48 h in total) at 37 °C. After each time-point, the plates were analyzed by CLSM as previously described.

### 5.18. Statistics and Reproducibility of Results

The data reported were statistically validated using the Student’s *t*-test comparing the mean absorbance of treated and untreated samples. The significance of differences between the mean absorbance values was calculated using a two-tailed Student’s *t*-test. A *p* < 0.05 was considered significant.

## Figures and Tables

**Figure 1 marinedrugs-20-00747-f001:**
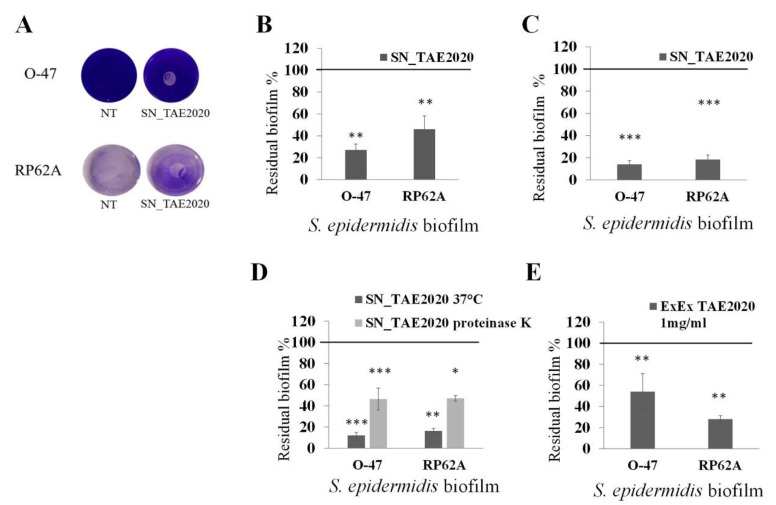
(**A**) Biofilm formation by *S. epidermidis* RP62A and *S. epidermidis* O-47 in polystyrene 24-wells microtiter plate wells coated with Gut medium (NT) and SN_TAE2020 obtained after 72 h of growth. (**B**) Effect of SN_TAE2020 on biofilm formation. (**C**) Effect of SN_TAE2020 on 24 h mature biofilm. (**D**) Effect of SN_TAE2020 treated with proteinase K on biofilm formation. (**E**) Effect of SN_TAE2020 organic extract (1 mg mL^−1^) on biofilm formation. Data are reported as the percentage of residual biofilm. Each data point represents the mean ± SD of six independent samples. The results are expressed as the percentage of biofilm formed in the presence of SN_TAE2020 or ExEx TAE2020 compared to untreated bacteria (100%). Biofilm formation was considered unaffected in the range of 90–100%. Differences in mean absorbance were compared to the untreated control and considered significant when *p* < 0.05 (* *p* < 0.05, ** *p* < 0.01, *** *p* < 0.001) according to the Student *t*-test.

**Figure 2 marinedrugs-20-00747-f002:**
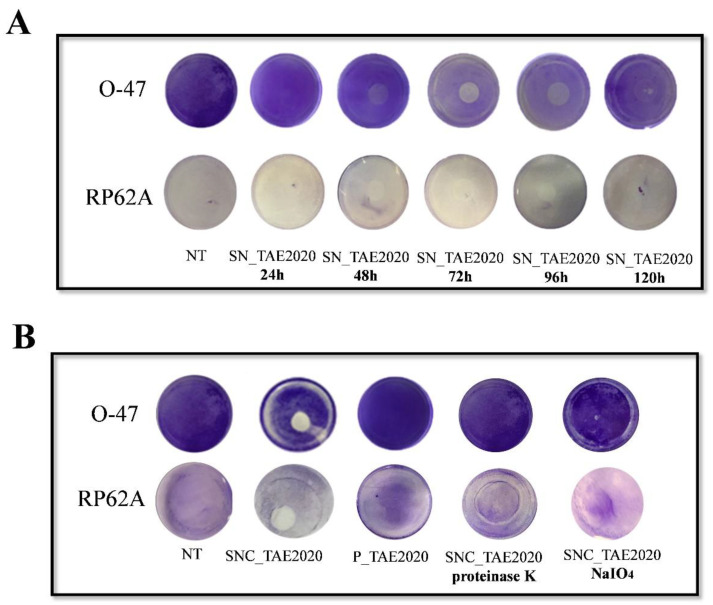
(**A**) Biofilm formation by *S. epidermidis* O-47 or *S. epidermidis* RP62A in polystyrene 24-wells microtiter plate wells coated with Gut medium (NT) and SN_TAE2020 obtained after different times of growth (24 h, 48 h, 72 h, 96 h, 120 h). (**B**) Biofilm formation by *S. epidermidis* O-47 or *S. epidermidis* RP62A in polystyrene 24-wells microtiter plate wells coated with Gut medium (NT), fractions obtained from the ultrafiltration process: SNC_TAE2020 (retentate fraction) and P_TAE2020 (permeate fraction) and SNC_TAE2020 treated with proteinase K or NaIO_4_.

**Figure 3 marinedrugs-20-00747-f003:**
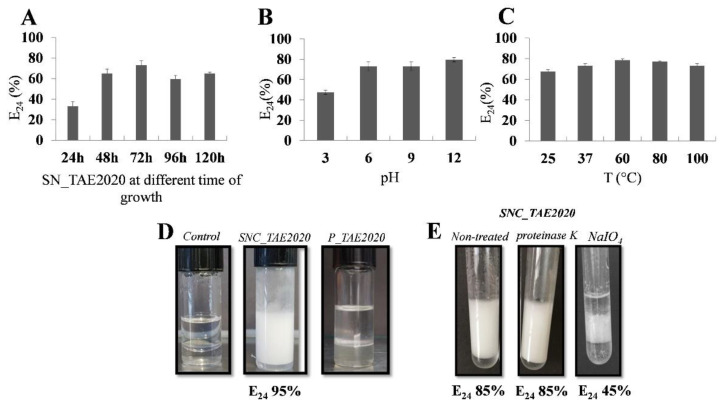
(**A**) Emulsification index (E_24_) of SN_TAE2020 obtained after different times of growth (24 h, 48 h, 72 h, 96 h, and 120 h); (**B**) Effect of pH on emulsification activity of SN_TAE2020; (**C**) Effect of temperature on emulsification activity of SN_TAE2020. The samples were heated at the indicated temperature for 30 min before the emulsification assay; (**D**) E_24_ of the fractions obtained from the ultrafiltration process (SNC_TAE2020 and P_TAE2020), Dectol emulsion with Gut medium is reported as control; (**E**) E_24_ of SNC_TAE2020 after the treatment with proteinase K and NaIO_4_, Dectol emulsion with SNC_TAE2020 non treated is reported as control. The solutions were mixed with Dectol (1:2 *v/v*) and analyzed after 24 h.

**Figure 4 marinedrugs-20-00747-f004:**
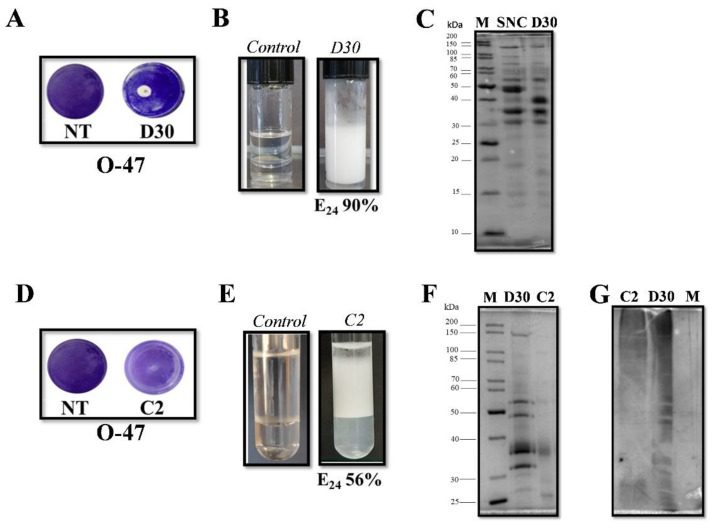
(**A**) Biofilm formation by *S. epidermidis* O-47 in polystyrene 24-wells microtiter plate wells coated with Gut medium (NT), and SNC_TAE2020 dialyzed against 10 volumes of Milli-Q water with a 30kDa PES (D30); (**B**) Emulsification index (E_24_) of D30 (**C**) SDS-PAGE 12,5% stained with Coomassie Blue, protein profile of SNC_TAE2020 and D30; (**D**) Biofilm formation by *S. epidermidis* O-47 in polystyrene 24-wells microtiter plate wells coated with Gut medium (NT), and the fraction C2; (**E**) Emulsification index (E_24_) of the C2 fraction; (**F**) Protein profile of fraction D30 and C2, M: molecular weight marker (DOC-PAGE 10% stained with Coomassie blue) (**G**) Polysaccharide analysis of fraction C2 and D30 (DOC-PAGE 10% stained with silver nitrate).

**Figure 5 marinedrugs-20-00747-f005:**
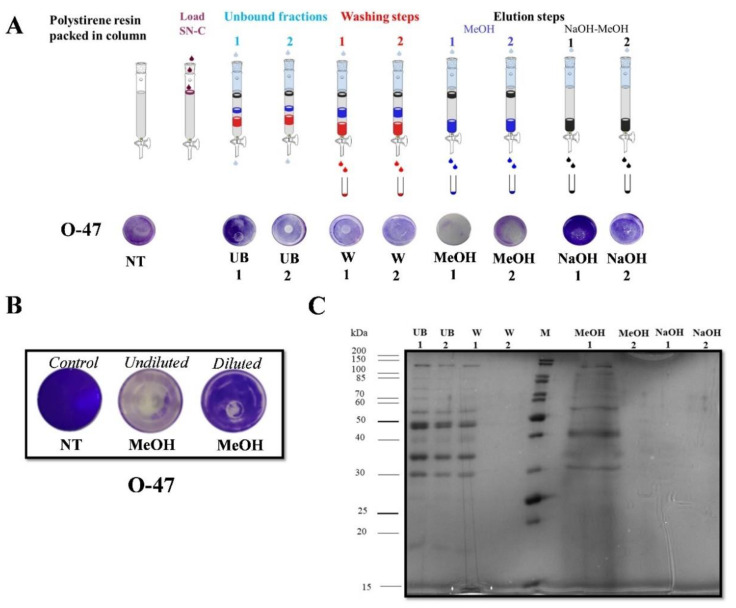
(**A**) Biofilm formation by *S. epidermidis* O-47 in polystyrene 24-wells microtiter plate wells coated with Gut medium (NT) and fractions obtained from adsorption chromatography: UB: unbound fractions, W: fractions obtained during the washing steps with Gut medium, MeOH: fractions eluted with methanol, NaOH: fractions eluted with NaOH in methanol; (**B**) Biofilm formation by *S. epidermidis* O-47 in polystyrene 24-wells microtiter plate wells coated with Gut medium (NT) with MeOH fraction and MeOH fraction diluted 4 times; (**C**) SDS-PAGE analysis of the main of the main chromatography fractions, M: molecular weight marker.

**Figure 6 marinedrugs-20-00747-f006:**
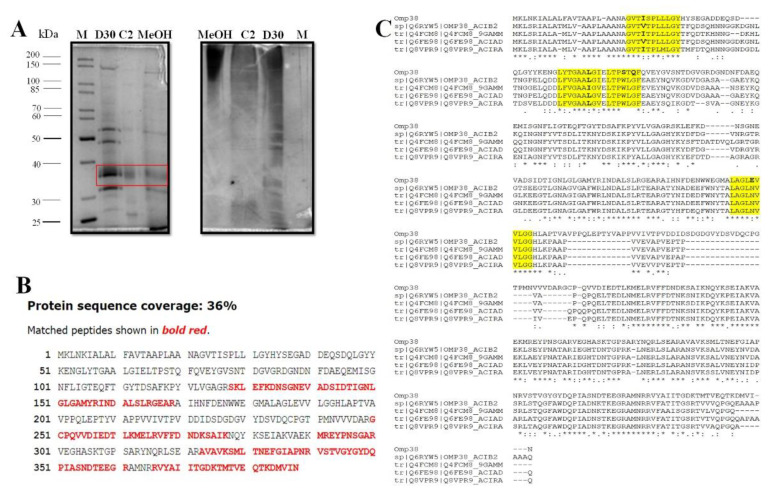
(**A**) Protein profile and polysaccharide detection of SNC_TAE2020 dialyzed against 10 volumes of Milli-Q water with a 30kDa PES (D30), fractions collected from the gel filtration process (C2), chromatography fraction eluted with methanol (MeOH), M: molecular weight marker (DOC-PAGE 10% stained with Comassie blu for protein visualization or silver nitrate for polysaccharide detection); (**B**) Protein sequence coverage obtained by LCMSMS analysis (**C**) Comparison of the amino acid sequences of *Psychrobacter sp.* TAE2020 Omp 38 and the OmpA of *Acinetobacter* Q8VPR9_ACIRA OmpA family protein OS = *Acinetobacter radioresistens*; Q4FCM8 Outer membrane protein A OS = *Acinetobacter sp.* V-26; Q6FE98 Putative Outer membrane protein (OmpA-like) OS = *Acinetobacter baylyi*; Q6RYW5|OMP38_ACIB2 Outer membrane protein Omp38 OS = *Acinetobacter baumannii*; hydrophobic loops were reported in yellow.

**Figure 7 marinedrugs-20-00747-f007:**
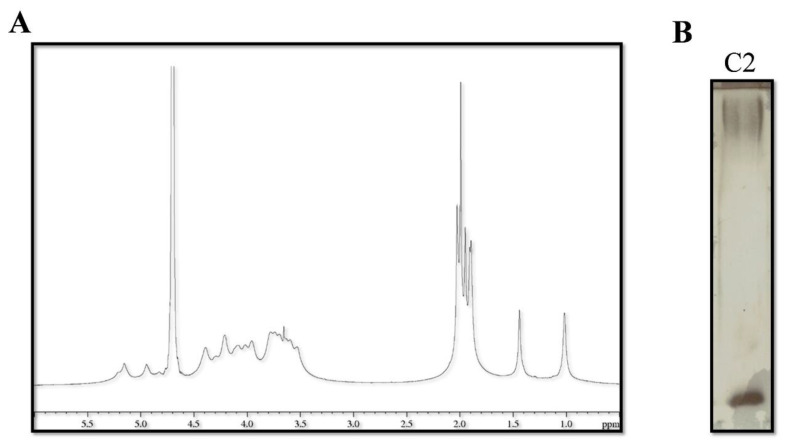
(**A**) ^1^H NMR spectrum of the fraction C2 in D2O; (**B**) DOC-PAGE analysis of the C2 fraction stained with Alcian Blue followed by silver nitrate.

**Figure 8 marinedrugs-20-00747-f008:**
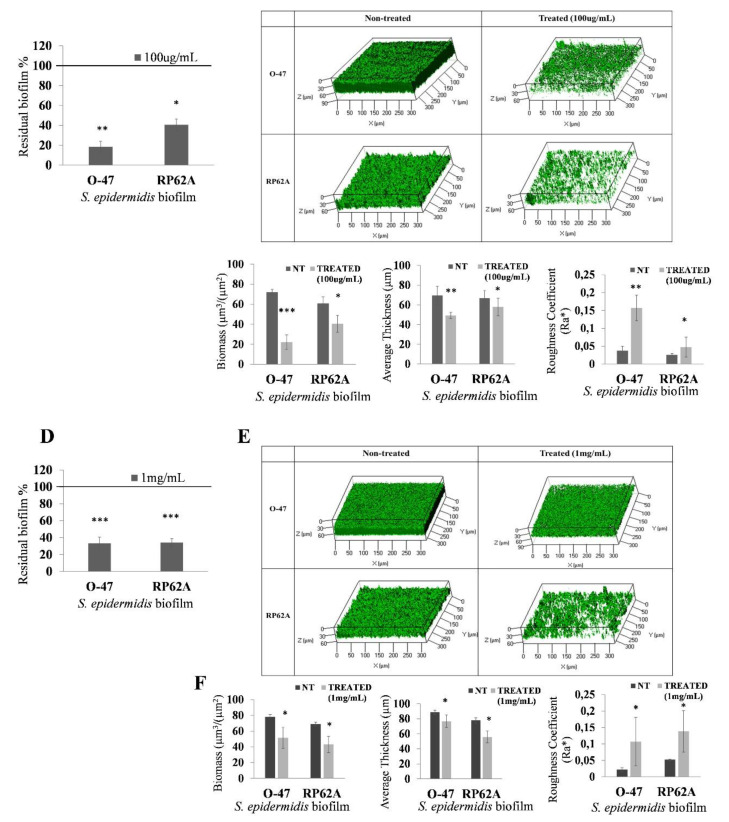
(**A**) Effect of CATASAN on *S. epidermidis* biofilm formation. Data are reported as the percentage of residual biofilm. Each data point represents the mean ± SD of six independent samples. The results are expressed as the percentage of biofilm formed in the presence of CATASAN compared to untreated bacteria (100%). Biofilm formation was considered unaffected in the range of 90–100%. (**B**) CLSM analysis of biofilms formed in the absence and presence (100 µg mL^−1^) of CATASAN. (**C**) COMSTAT quantitative analysis of biomass and average thickness of treated (CATASAN) and untreated (NT) biofilms. (**D**) Effect of CATASAN (1 mg mL^−1^) on *S. epidermidis* 24 h mature biofilm. Data are reported as the percentage of residual biofilm. Each data point represents the mean ± SD of six independent samples. The results are expressed as the percentage of biofilm formed in the presence of CATASAN compared to untreated bacteria (100%). Biofilm formation was considered unaffected in the range of 90–100%. (**E**) CLSM analysis of 24 h mature biofilm formed in the absence and presence (1 mg mL^−1^) of CATASAN. (**F**) COMSTAT quantitative analysis of biomass and average thickness of treated (CATASAN) and untreated (NT) biofilms. Three-dimensional biofilm structures were obtained using the LIVE/DEAD^®^ Biofilm Viability Kit. Differences in mean absorbance were compared to the untreated control and considered significant when *p* < 0.05 (* *p* < 0.05, ** *p* < 0.01, *** *p* < 0.001) according to the Student *t*-test.

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
