# Peer review of "CATASAN Is a New Anti-Biofilm Agent Produced by the Marine Antarctic Bacterium *Psychrobacter* sp. TAE2020"

_marinedrugs, 2022, doi:10.3390/md20120747_

Round 1

Reviewer 1 Report

The submitted manuscript is an interesting study containing many valuable data. However, I would like to indicate several comments, which can help to improve the article:

1.       Several editing errors (lacking dot and break in line 75, lacking italic in line 95, etc.)

2.       The scientific novelty of the results presented in chapter 2.1. is disputable in the light of cited articles [23-25] mentioned in lines 88-91. Please indicate why this experiment has been conducted?

3.       Line 168: “Emulsifying activity increased after heating” – why? What mechanism do you suggest? What about thermal degradation of tested compounds?

4.       Line 182: Why were the samples treated with proteinase K or NaIO4? Please explain in the manuscript.

5.       Line 187: The conclusion is that the bioemulsifying properties are caused by carbohydrates. However, the following chapters describe protein analysis. Please explain.

6.       Fig. 5C. Please provide bigger image with better resolution.

7.       Could you propose the possible molecular structure of CATASAN?

8.       In “Results” section please provide information about statistical analysis of the results.

Author Response

We are deeply obliged to reviewer 1 for the careful evaluation of the manuscript and his/her valuable suggestions. Following the referees’ suggestions, the text and figures were modified. We hope that the revised version is now acceptable for publication in Marine Drugs.

Reviewer: 1
The submitted manuscript is an interesting study containing many valuable data. However, I would like to indicate several comments, which can help to improve the article:

  1. Several editing errors (lacking dot and break in line 75, lacking italic in line 95, etc.)

Re: We thank the Referee and corrected the typos

  1. The scientific novelty of the results presented in chapter 2.1. is disputable in the light of cited articles [23-25] mentioned in lines 88-91. Please indicate why this experiment has been conducted?

Re: The articles [23-25] describe the isolation and the genome sequencing of the two S.epidermids strains used as target strains, the anti-adhesive and anti-biofilm activity of SN_TAE2020 on the biofilm of these two strains are new results never described before. We thank the referee for the comment, and we modified the sentence to clarify this point (lines 84-88).

  1. Line 168: “Emulsifying activity increased after heating” – why? What mechanism do you suggest? What about thermal degradation of tested compounds?

Re: We really thank the Referee for this comment, indeed thanks to his/her comment we discovered that we made a not correct comment about these results indeed the emulsifying activity is stable at high temperatures it didn’t increase after hearing. We accordingly modified the text (lines 168-169).

  1. Line 182: Why were the samples treated with proteinase K or NaIO4? Please explain in the manuscript.

Re: According to the Referee’s suggestion, we introduced a sentence to better explain why we used proteinase K or NaIO4 (lines 182-185)

  1. Line 187: The conclusion is that the bioemulsifying properties are caused by carbohydrates. However, the following chapters describe protein analysis. Please explain.

Re: Although the treatment with NaIO4 suggested that the bioemusifing activity is related to carbohydrates we are also interested in anti-adhesive activity therefore we set up two different purification protocols; one inspired by the protocols designed to purify emulsifiers produced by marine bacteria and one based on the ability of our anti-adhesive molecule to bind polystyrene. For both purification protocols, each function was evaluated for anti-adhesive and emulsifying activity, and for each fraction the protein and polysaccharidic profiles were analyzed.

  1. 5C. Please provide bigger image with better resolution.

Re: The dimension and resolution of the figures were improved.

  1. Could you propose the possible molecular structure of CATASAN?

Re: We are extremely interested in the molecular structure of Catasan but we need a punctual structure of the polysaccharide component of CATASAN to figure out a possible molecular structure and to characterize the interaction between the polypeptidic chain and polysaccharides. The definition of the polysaccharide structure is a long and sometimes complex experimental work, but we are working on it.

  1. In “Results” section please provide information about statistical analysis of the results.
    Re: We are deeply indebted to Reviewer 1 for having raised this point, in the previous version of the manuscript we missed providing the information on statistical analysis, we introduced the statistical analysis details in paragraph 5.18 and in each figure legend.

Reviewer 2 Report

The current MS entitled ``CATASAN: A new anti-biofilm agent produced by the marine Antarctic bacterium Psychrobacter sp. TAE2020`` can not be accepted with respect to Marine Drugs. All the raised issues below should be carefully addressed.

-Extensive English editing is needed there are many grammatical and typing mistakes

-The bacteria name should be italicized in the title and in all MS.

-- the abstract needs rewriting to be more informative. 

-the tools for identification should be mentioned in the abstract of this protein polysaccharide 

-Methods of assays and some values of this compound along with positive control should be added.

-key words should be correct as follows

-Revise the first sentence in the introduction.

-For lines 35 to 37, a reference should be added.

-In the introduction authors should add more about this bacteria. Are there any metabolites previously reported from this bacteria?

-Results section 

 lines from 82 to 100, this part belongs to materials and methods because they mentioned the techniques, methods 

 - All microorganisms' names should be italicized throughout the whole MS.

 - All the figures need improvement as they are unreadable.

 - A list of abbreviations should be provided.

 - Can authors explain (data not shown), why they didn't not all obtain data in this work?

 -in page 6' line 2004, a reference should be added.

 -All the results sections should be revised and omit the parts related to materials and methods sections and include them in their proper location in the MS.

 -there are so many repetitions, the MS should be carefully revised and omit the repeating parts. 

 -The references should be cited probably, [46] [47] [39], check this issue through the whole MS.

 -Conclusion is missing.

Author Response

We are deeply obliged to  reviewer 2 for the careful evaluation of the manuscript and his/her valuable suggestions. Following the referee’s suggestions, the text and figures were modified. We hope that the revised version is now acceptable for publication in Marine Drugs.

Comments and Suggestions for Authors

The current MS entitled ``CATASAN: A new anti-biofilm agent produced by the marine Antarctic bacterium Psychrobacter sp. TAE2020`` can not be accepted with respect to Marine Drugs. All the raised issues below should be carefully addressed.

Re: The title of the paper was modified accordingly with the referee’s request and following the Marine drugs author’s instruction (The title of your manuscript should be concise, specific, and relevant. It should identify if the study reports (human or animal) trial data, or is a systematic review, meta-analysis or replication study. When gene or protein names are included, the abbreviated name rather than full name should be used. Please do not include abbreviated or short forms of the title, such as a running title or head). We thank the reviewer since the new title is clearer and helps us to introduce the name of the newly discovered complex.

-Extensive English editing is needed there are many grammatical and typing mistakes

Re: The manuscript was carefully edited by an outside party

-The bacteria name should be italicized in the title and in all MS.

Re: We thank the Referee, and we corrected accordingly the text

-- the abstract needs rewriting to be more informative.

Re: The abstract was improved to be more informative.

-the tools for identification should be mentioned in the abstract of this protein polysaccharide 

Re: According to the Referee’s suggestion the abstract was modified to introduce more information and details about the tools used for the identification of protein and polysaccharide components of CATASAN.

-Methods of assays and some values of this compound along with positive control should be added.

Re: In biofilm assays, the data are reported as the percentage of residual biofilm because the biofilm biomass change in different conditions, and for the two S.epidermidis strains, for example, the absorbance value of untreated mature biofilm in the case of S.epidermidis RP62A is 0.5 OD590nm, while in the case of S. epidermidis O47 the value is 3OD590nm .

-key words should be correct as follows

Re: As requested we modified the Keywords

-Revise the first sentence in the introduction.

Re: According to the Referee’s suggestion the sentence was revised.

-For lines 35 to 37, a reference should be added.

Re: As requested we introduced a reference.

-In the introduction authors should add more about this bacteria. Are there any metabolites previously reported from this bacteria?

Re: We started to work on this bacterium very recently and only two papers were published on the features of this strain, we reported all data available in the introduction section.

-Results section 

 lines from 82 to 100, this part belongs to materials and methods because they mentioned the techniques, methods

Re: According to the Referee’s suggestion, we removed the technical details.

 - All microorganisms' names should be italicized throughout the whole MS.

Re: We thank the Referee and corrected the typos

 - All the figures need improvement as they are unreadable.

Re: The dimension and resolution of the figures were improved.

 - A list of abbreviations should be provided.

Re: We really thank the Referee for this suggestion, and we introduced a list of abbreviations

 - Can authors explain (data not shown), why they didn't not all obtain data in this work?

Re: in the manuscript, some results were not shown to avoid the introduction of not necessary images as in the case of emulsion pictures:

  • It is remarkable to note that the emulsion is stable for more than 30 days at room temperature (data not shown).

  • These data demonstrated that SNC_TAE2020 treated with proteinase K didn’t lose emulsifying capability, although the treated sample formed less stable emulsification (data not shown)

  • The sample D30 was fractionated by a gel filtration chromatography and all the collected fractions were assayed for coating and emulsifying activity (data not shown).

Or in the case of biofilm assays performed on both S.epidermidis stains that have the same results:

  • The results showed that the fractions eluted with methanol had strong anti-adhesive activity against both staphylococcal strains (data not shown).

Or in the case of BLAST analysis

  • BLAST analysis search revealed that this protein is similar to OmpA family proteins from different Psychrobacter strains (data not shown).

 -in page 6' line 2004, a reference should be added.

Re: we accordingly modified the text

 -All the results sections should be revised and omit the parts related to materials and methods sections and include them in their proper location in the MS.

Re: The text was modified as suggested.

 -there are so many repetitions, the MS should be carefully revised and omit the repeating parts. 

Re: We are deeply indebted to the Reviewer for having raised this point the paper was carefully revised and we hope that in the new version it is clear

 -The references should be cited probably, [46] [47] [39], check this issue through the whole MS.

Re: the references were cited in line 439--..

 -Conclusion is missing.

Re: A paragraph containing the conclusion was added.

Round 2

Reviewer 1 Report

Thank the Authors for theeir kind responses. 

Reviewer 2 Report

No comment